# ZP4 confers structural properties to the zona pellucida essential for embryo development

Ismael Lamas-Toranzo[1], Noelia Fonseca Balvís[1], Ana Querejeta-Fernández[2], María José Izquierdo-Rico[3], Leopoldo González-Brusi[3], Pedro L Lorenzo[4], Pilar García-Rebollar[5], Manuel Avilés[3]*, Pablo Bermejo-Álvarez[1]*

[1]Animal Reproduction Department, INIA, Madrid, Spain; [2]Department of Physical Chemistry and Biomedical Research Center (CINBIO), Universidad de Vigo, Vigo, Spain; [3]Cell Biology and Histology Department, Faculty of Medicine, Universidad de Murcia and IMIB-Arrixaca, Murcia, Spain; [4]Animal Physiology Department, Veterinary Faculty, Universidad Complutense de Madrid, Madrid, Spain; [5]Animal Production Department, ETSI Agrónomos, Universidad Politécnica de Madrid, Madrid, Spain

**Abstract** Zona pellucida (ZP), the extracellular matrix sheltering mammalian oocytes and embryos, is composed by 3 to 4 proteins. The roles of the three proteins present in mice have been elucidated by KO models, but the function of the fourth component (ZP4), present in all other eutherian mammals studied so far, has remained elusive. Herein, we report that *ZP4* ablation impairs fertility in female rabbits. Ovulation, fertilization and in vitro development to blastocyst were not affected by *ZP4* ablation. However, in vivo development is severely impaired in embryos covered by a ZP4-devoided zona, suggesting a defective ZP protective capacity in the absence of ZP4. ZP4-null ZP was significantly thinner, more permeable, and exhibited a more disorganized and fenestrated structure. The evolutionary conservation of ZP4 in other mammals, including humans, suggests that the structural properties conferred by this protein are required to ensure proper embryo sheltering during in vivo preimplantation development.
DOI: https://doi.org/10.7554/eLife.48904.001

*For correspondence:
maviles@um.es (MA);
bermejo.pablo@inia.es (PB-Á)

**Competing interests:** The authors declare that no competing interests exist.

## Introduction

Mammalian oocytes and early preimplantation embryos are surrounded by a glycoprotein shell termed zona pellucida (ZP) that serves important functions during folliculogenesis, species-specific fertilization and early development. In eutherian mammals, this glycoprotein shell is solely composed of 3 to 4 proteins depending on the species. The differences in ZP composition between eutherian mammals is the consequence of duplication and/or pseudogenization events during evolution. In all eutherian mammals studied so far ZP always contain ZP2 and ZP3 proteins, but ZP1 or ZP4 may be absent, as both are paralogous genes formed by duplication of a common ancestry gene (reviewed by *Goudet et al., 2008*).

The functions of the three proteins (ZP1, ZP2 and ZP3) present in mouse ZP have been elucidated by gene ablation experiments. ZP1 serves a structural function, being dispensable for sperm binding or fertilization (*Rankin et al., 1999*). ZP1 null females display perturbed folliculogenesis and, although ovulation rates are not affected, both cleavage rate and litter size are reduced, likely due to ZP1-null zona being unable to protect the developing embryo. ZP2 ablation also results in structural defects, which are more severe, as the number of antral follicles is reduced and only few ZP-free oocytes and no two-cell embryos are recovered from *Zp2* knock-out (KO) females after mating

**eLife digest** The egg cells of mammals, called oocytes, are encased in a protective layer called the zona pellucida. This layer is made from proteins called ZP1 to 4. Most studies of the zona pellucida use mice, which do not have ZP4. This means that the research community have limited knowledge of what ZP4 does in humans and other mammals.

Scientists can now use a technique called CRISPR to selectively modify the genetics of living things to help us to understand what specific genes and proteins do. The ZP4 protein can be eliminated from rabbit oocytes using CRISPR to help understand its role in egg cell fertilization and development.

Lamas-Toranzo et al. examined the effect of losing ZP4 from rabbit oocytes. Without ZP4 the zona pellucida becomes thinner, irregular and more flexible. However, the loss of ZP4 did not affect ovulation (i.e. the release of egg cells from an ovary), fertilization, or the early stages of development of embryos when studied in the laboratory. However, rabbits without ZP4 were much less fertile. Indeed, only one out of 10 female rabbits without ZP4 was able to deliver pups because in most cases the development of embryos in the womb failed.

These findings show that ZP4 has a structural role in the zona pellucida. Without ZP4 fertility is reduced. This work lays the ground for further investigation of the role of ZP4. It could also offer new insights into the causes of infertility.

DOI: https://doi.org/10.7554/eLife.48904.002

(*Rankin et al., 2001*). Finally, *Zp3* KO mice are unable to form ZP, showing a drastic reduction in ovulation rates and also being unable to produce cleaved embryos after mating (*Liu et al., 1996*; *Rankin et al., 1996*). Given the lack of ZP on ovulated *Zp2*- or *Zp3*-null oocytes, these KO models could not assess the role of these proteins on sperm binding. To study that process, more complex gain-of-function transgenic mouse models have been generated. In particular, the generation of ZP composed of different combinations of human ZP proteins (*Baibakov et al., 2012*) or an heterologous ZP lacking mouse ZP2 protein (*Avella et al., 2014*) have uncovered a crucial role of ZP2 on mouse and human sperm recognition.

The fourth ZP protein component, ZP4, is absent in mice, which seems to be an exception for mammals, as ZP4 is present in multiple mammalian species including humans (*Hughes and Barratt, 1999*; *Lefièvre et al., 2004*), rabbits (*Stetson et al., 2012*), ungulates (*Hedrick and Wardrip, 1987*; *Noguchi et al., 1994*; *Topper et al., 1997*), carnivores (*Moros-Nicolás et al., 2018*) and other rodents (*Hoodbhoy et al., 2005*; *Izquierdo-Rico et al., 2009*). The lack of ZP4 in laboratory mouse (*Mus musculus*), the only species where KO models were readily available, has precluded the study of the role of this protein by of loss-of-function experiments. Gene ablation experiments are essential to unequivocally know the function of a gene, but were largely restricted to mice, given the technical difficulties to perform site-specific genome modification in other mammalian species (*Lamas-Toranzo et al., 2017*). Luckily, the advent of site-specific endonucleases, and particularly CRISPR, the last technology to be developed and the easiest to tailor to its genomic target, allows direct KO generation in one step. Using this technology, we have generated *ZP4*-null rabbits uncovering that ZP4 confers structural properties to the ZP that are essential to protect the developing preimplantation embryo.

## Results

### Generation of *ZP4* KO rabbits

*ZP4* gene ablation was achieved by CRISPR-mediated mutagenesis at the zygote stage. For this aim, a sgRNA targeting the beginning of *ZP4* coding region was designed (*Figure 1A*). Cas9-coding capped polyadenylated mRNA and *ZP4*-targeting sgRNA were injected into the ooplasm at 100 and 25 ng/µl, respectively. Following microinjection, 50 embryos were transferred to the oviduct of two pseudopregnant recipients (20 and 30 embryos/female), resulting in the delivery of 12 pups (5 and 7, respectively). Genotyping was performed on ear biopsies and blood samples from each pup. Genome edition rates were assessed by sequencing a PCR product containing the target site. This

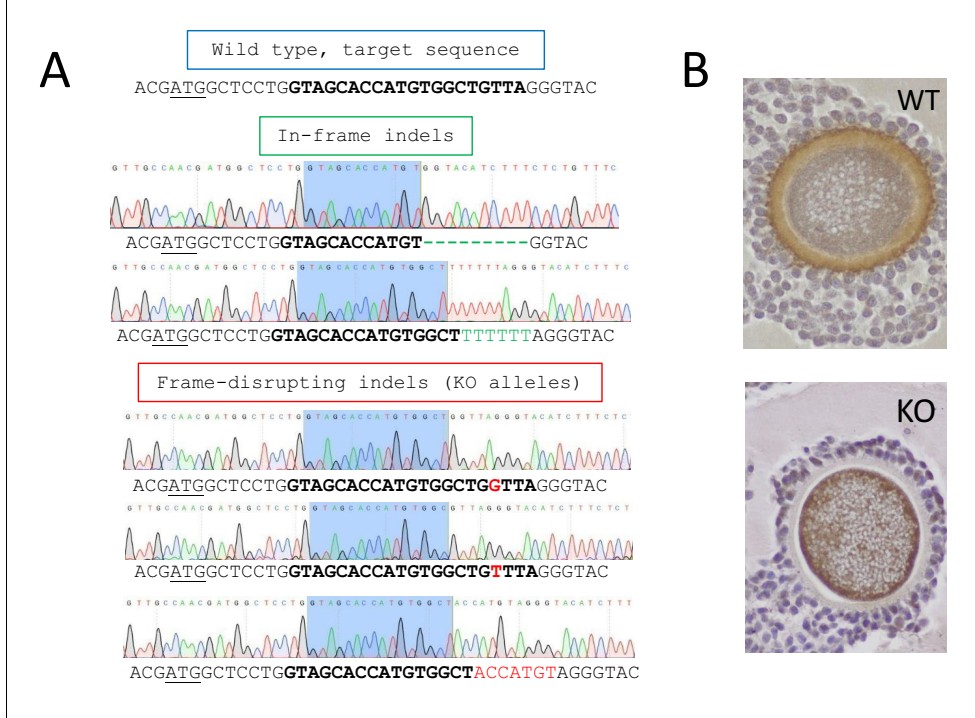

**Figure 1.** Generation of *ZP4* KO rabbits. (**A**) A sgRNA was designed for a genomic target located at the beginning of the coding region. Upper sequence depicts wild-type sequence with the start codon underlined and CRISPR target sequence marked in bold letters. Examples of different indels obtained in F0 generation are depicted below. Both in-frame indels (i.e., multiple of three) and frame-disrupting (i.e., not multiple of three) were obtained. Only frame disrupting indels generate truly KO alleles by disrupting the ORF of *ZP4*. The KO allele involving a G insertion was selected to establish a line. (**B**) Wild-type and *ZP4*-null cumulus oocyte complexes following immunohistochemistry with an anti-ZP4 antibody.

DOI: https://doi.org/10.7554/eLife.48904.003

The following figure supplement is available for figure 1:

**Figure supplement 1.** Sequencing of the five most probable off-target sequences, revealing no off-target genome editing.

DOI: https://doi.org/10.7554/eLife.48904.004

---

initial screening revealed that all 12 pups were edited at the target site, but it did not allow to identify each mutated allele harbored by each individual. For this aim, clonal sequencing was performed, revealing that all pups but one male were mosaic, that is contained more than two alleles. This phenomenon appears when DNA replication precedes genome edition and has been commonly observed following zygote microinjection in different mammalian species (*Lamas-Toranzo et al., 2017*), including rabbits (*Guo et al., 2016*; *Honda et al., 2015*; *Lv et al., 2016*; *Song et al., 2016a*; *Song et al., 2016b*; *Sui et al., 2016*; *Yan et al., 2014*; *Yang et al., 2016*; *Yuan et al., 2016*), where genome replication takes place shortly after fertilization (*Oprescu et al., 1965*; *Szollosi, 1966*).

In the absence of an homologous recombination template, CRISPR-mediated gene ablation relies on the generation of frame-disrupting insertion-deletions (indels) by non-homologous end joining (NHEJ) (*Lamas-Toranzo et al., 2017*). As some of the indels generated do not alter the Open Reading Frame of the gene, only those individuals harboring frame-disrupting mutations in all alleles can be considered KO. None of the seven females generated in F0 were KO, as all contained at least one in-frame indel (*Figure 1A*). This situation precluded the use of F0 generation for direct experimentation. Consequently, founders were breed to obtain wild-type (WT), heterozygous (Hz) or knock-out (KO) females in subsequent generations. A non-mosaic male containing two frame-disrupting indels was able to breed normally, as expected, given that ZP4 protein is exclusively expressed in ovaries. This male was crossed to mosaic females to generate KO animals (i.e., harboring two frame-disrupting indels) that were used for an initial fertility (pregnancy) screening.

Subsequently, one of the KO alleles carried by that male and formed by a G insertion was selected in Hz (WT/KO) animals to maintain the line and generate WT, Hz and KO individuals for experiments. Off-target analysis revealed no genome edition in any of the five most probable off-target sites (*Figure 1—figure supplement 1*). Protein ablation in KO female individuals was confirmed by IHC in ovaries (*Figure 1B*). Proteomics analysis of solubilized extracts of ZP from oocytes obtained from KO and WT individuals by tandem mass spectrometry further confirmed the absence of ZP4 in KO individuals. *ZP4* ablation did not prevent the synthesis of all other three rabbit ZP proteins (ZP1-3), which were also detected in the solubilized extract of *ZP4* KO ZP (*Supplementary file 1*). This analysis also provides a semi-quantitative estimation of the amount of ZP1-3 proteins based on the number of peptides detected for each protein, which were similar for KO and WT oocytes, thereby suggesting that ZP4 ablation does not induce major alterations in ZP1-3 synthesis.

## ZP4 ablation leads to severely impaired fertility

The effect of *ZP4* ablation on female fertility was initially tested by crossing WT, Hz or KO females with fertile WT males. WT and Hz females delivered normal litter sizes, whereas only one out of 10 KO females, delivered a single litter of reduced size (four pups, *Figure 2A*). In order to elucidate the root for the reproductive failure caused by *ZP4* ablation, ovulation and embryonic cleavage rates were assessed in WT, Hz or KO females. For this aim, five females per group were crossed with WT males and sacrificed 15 hr after mating. Ovulation rates following natural mating were similar between all groups, suggesting that *ZP4* ablation did not impair folliculogenesis (*Figure 2B*). To further test the normalcy of *ZP4*-null follicles, a histological analysis was performed on KO or WT ovaries. *ZP4* disrupted ovaries were grossly indistinguishable from WT or Hz and no obvious abnormalities were noted in follicular population (*Figure 2E–H* and *Figure 2—figure supplement 1*).

To analyse embryonic cleavage rates, presumptive zygotes were obtained by oviductal flushing and subsequently cultured in vitro. Cleavage rates did not differ between the different groups, suggesting that ZP4 does not play an essential role in fertilization in vivo (*Figure 2C*). Cleaved embryos were allowed to develop to the blastocyst stage in vitro. Again, no differences were noted in blastocyst rates or morphology, showing that embryos derived from *ZP4*-null oocytes (Hz embryos enclosed by a ZP4-null ZP, MATKO embryos) are able to reach the blastocyst stage in vitro (*Figure 2D*). To further confirm that spermatozoa were able to penetrate through the *ZP4*-disrupted zona, pronuclei formation was assessed in zygotes collected from KO and WT (three animals per group). Normal pronuclear formation was noted for embryos derived from *ZP4*-null oocytes, showing no effect of *ZP4* ablation on the in vivo fertilization process (*Figure 2G*). As expected, given that rabbit ZP is not involved in polyspermy blockage (*Pincus and Enzmann, 1932*), no polyspermy was found on embryos produced from *ZP4*-null oocytes.

As ovulation and fertilization were not affected in *ZP4*-null females, the negative effect of *ZP4* ablation can only be attributed to a developmental failure of the embryos surrounded by a ZP lacking ZP4. However, given that these Hz embryos derived from *ZP4*-disrupted oocytes were able to develop in vitro to the blastocyst stage, this developmental failure must occur in vivo. ZP protection is essential for in vivo embryo development, as rabbit embryos without zona are unable to establish pregnancy following embryo transfer (*Moore et al., 1968*; *Rottmann and Lampeter, 1981*). In vivo development was analysed in KO and WT females (three per group) crossed with fertile males and sacrificed on Day 4 post-mating. At this developmental time, rabbit embryos have developed to the blastocyst stage and are surrounded by a mucin coat, but lack ZP, which disappears between Days 3 and 4 (*Denker and Gerdes, 1979*; *Fischer et al., 1991*). ZP dissolution does not occur in vitro as it requires both from embryo development and uterine secretions (*Fischer et al., 1991*). All embryos recovered from WT females at Day 4 post-mating had reached the blastocyst stage, however, less than half of the MATKO embryos recovered from KO females were able to develop to the blastocyst stage (*Figure 3A*). Moreover, those embryos reaching the blastocyst stage in the KO group showed a significant reduction in blastocyst expansion (~500 vs.~200 μm diameter for WT vs. KO, respectively, *Figure 3B*). ZP was absent in all WT blastocysts (*Figure 3C*) and in most expanded or collapsed blastocysts from the KO group, but was present in all degenerated embryos (*Figure 3D*), in agreement with the requirement of embryo development for ZP dissolution previously reported (*Fischer et al., 1991*). Evident signs of mechanical damage (non-spherical, crushed ZP) could be

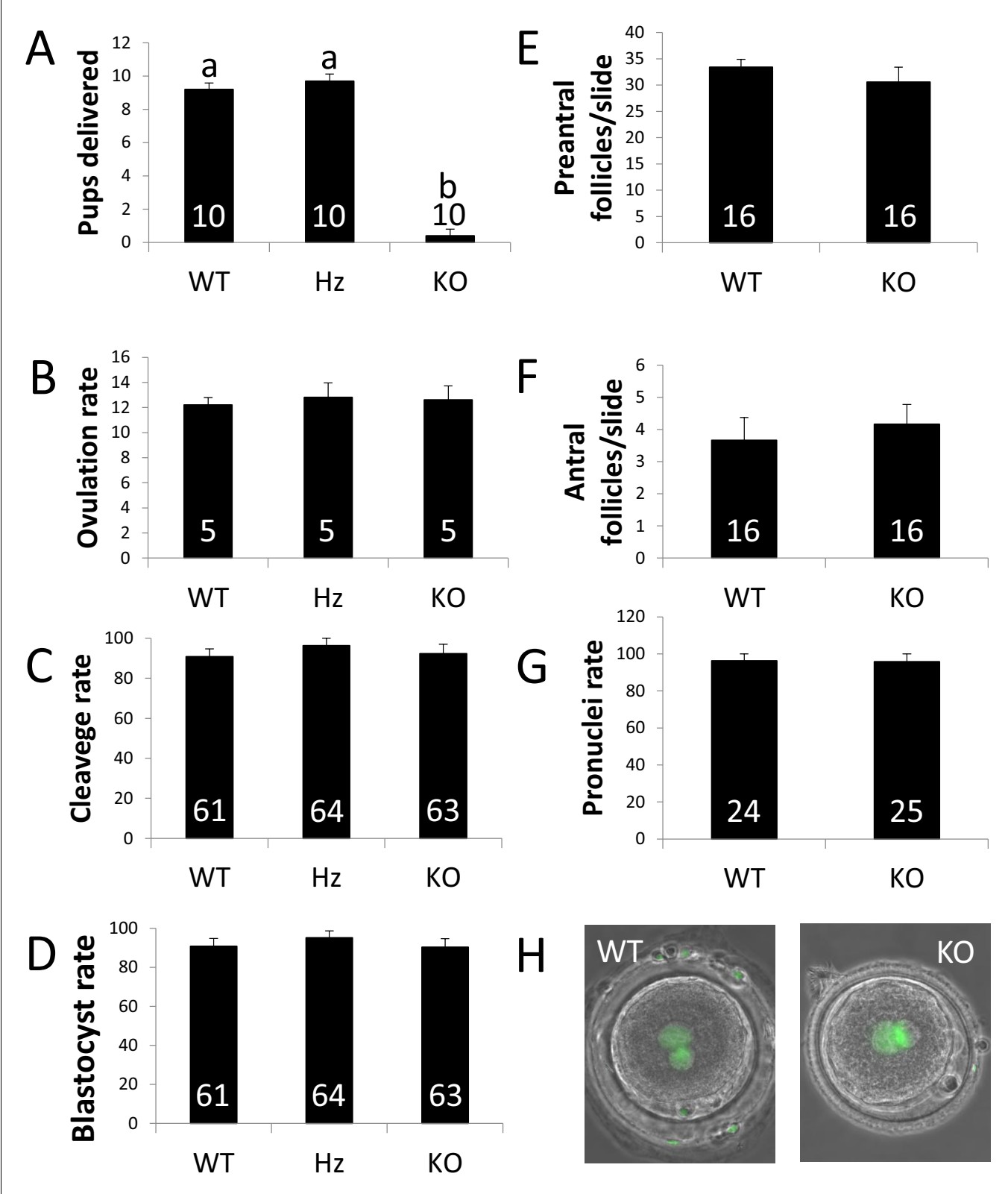

**Figure 2.** ZP4 ablation leads to severely impaired fertility. (**A**) Number of pups delivered in WT, Hz and KO females. *ZP4* ablation significantly impaired fertility: only one female out of 10 delivered a small litter of 4 pups, whereas *ZP4* haploinsufficiency (Hz) did not affect fertility. (**B–G**) *ZP4* disruption did not affect ovulation (**B**), cleavage (**C**) and blastocyst (**D**) rates, preantral (**E**) and antral (**F**) follicles counts or pronuclei formation rate (**G**). The number of females (**A–B**), embryos (**C-D** and **G**) and slides (**E–F**) analysed per group is indicated inside each column. Data are represented as mean ± SEM in all

*Figure 2 continued on next page*

*Figure 2 continued*

graphs. Statistical differences are based on ANOVA (p<0.05, (A–D) or t-test (p<0.05, (E–F). (H) Representative images of zygotes obtained from WT or KO females showing normal pronuclear formation (SYTOX green staining) in the presence (WT) or absence (KO) of ZP4 in ZP.

DOI: https://doi.org/10.7554/eLife.48904.005

The following figure supplement is available for figure 2:

**Figure supplement 1.** ZP4 ablation does not cause significant alterations in folliculogenesis.

DOI: https://doi.org/10.7554/eLife.48904.006

observed in degenerated embryos in the KO group (*Figure 3D*), suggesting that mechanical pressure may be at least partially responsible for the developmental failure.

Beyond Day 4 and before implantation, the rabbit embryo undergo a relevant embryonic growth that has not been recapitulated yet on an in vitro setting. This growth, converting a ~ 500 µm blastocyst to a ~ 3 mm expanded blastocysts from Days 4 to 6, is concomitant to the formation of a new glycoprotein matrix termed neozona and the replacement of the mucin coat by the gloiolemma (*Fischer et al., 1991*). To determine the carry over effects of the developmental delay observed on Day 4, in vivo development was analysed in KO and WT females (four per group) crossed with fertile males and sacrificed on Day 6 post-mating. Following post-mortem recovery, we observed that, based on the number of corpora lutea of each individual, more than half of the MATKO embryos (Hz embryos lacking ZP4 in their ZP) had been already lost by Day 6 after mating (*Figure 4A*), which roughly coincide with the percentage of degenerated embryos on Day 4. Furthermore, the few embryos surrounded by ZP4-null ZP present in the uterus that were not degenerated by Day 6 were developmentally arrested compared to those protected by WT ZP, as blastocyst growth was severely impaired (*Figure 4B–E*). These results suggest that *ZP4* confers structural properties to the ZP that are required for embryo protection during preimplantation development in vivo.

## ZP4-devoided zona show structural abnormalities

Morphological differences between KO and WT ZP were clearly noticeable under light microscopy: ZP lacking ZP4 appeared thinner and more irregular (less spherical) compared to WT ZP (*Figure 5A*). Furthermore, although a quantifiable mechanical analysis was not performed, ZP4-devoided ZP appeared easily deformable under the mechanical pressure exerted by a blunt micromanipulation needle compared to WT ZP (*Video 1*). In regard to ZP thickness, measurements by contrast light microscopy revealed a significant reduction (~4 µm) in the absence of ZP4 to the ZP of WT or Hz females (*Figure 5A*). No differences in ZP thickness were noticed between WT or Hz females, suggesting that *ZP4* haploinsufficiency does not produce any obvious alteration in ZP formation, in agreement with the lack of differences in fertility between WT and Hz individuals. To further characterize the effect of *ZP4* ablation on ZP texture, we analysed WT or KO ZP from in vivo derived zygotes (~14–15 hr post-mating) by Scanning Electron Microscopy (SEM). SEM images evidenced notable differences in textural properties between WT and ZP4 lacking ZP: while the former exhibited a smooth and compact texture, the latter displayed an uneven, rougher and porous aspect (*Figure 5B*). This less compact structural organization can explain its higher deformability and the larger fenestrations could lead to increased permeability. ZP permeability was tested by incubating zygotes obtained from WT or *ZP4*-null females with green fluorescent nanospheres of different diameters, ranging from 20 to 40 and 100 nm. This analysis revealed that, while 100 nm-sized nanospheres were successfully blocked by both ZPs, all 9 ZPs lacking ZP4 analyzed were permeable to 20 to 40 nm nanospheres, which are efficiently blocked by all 9 WT ZPs (*Figure 5C*).

## Discussion

Despite its essential functions during folliculogenesis, fertilization and preimplantation embryo development, ZP is composed of just 3 to 4 proteins, depending on the species. Gene ablation experiments have deciphered the functions of the three proteins present in mouse ZP, but the role of the fourth component (ZP4) has remained controversial. Previous studies have suggested that ZP4 may play a role during fertilization. Rabbit ZP4 was found to bind to rabbit sperm acrosome (*Prasad et al., 1996*), although specific blocking by ZP4-antisera fragments could not be demonstrated. Similarly, other studies based on in vitro protein-binding assays have attributed sperm

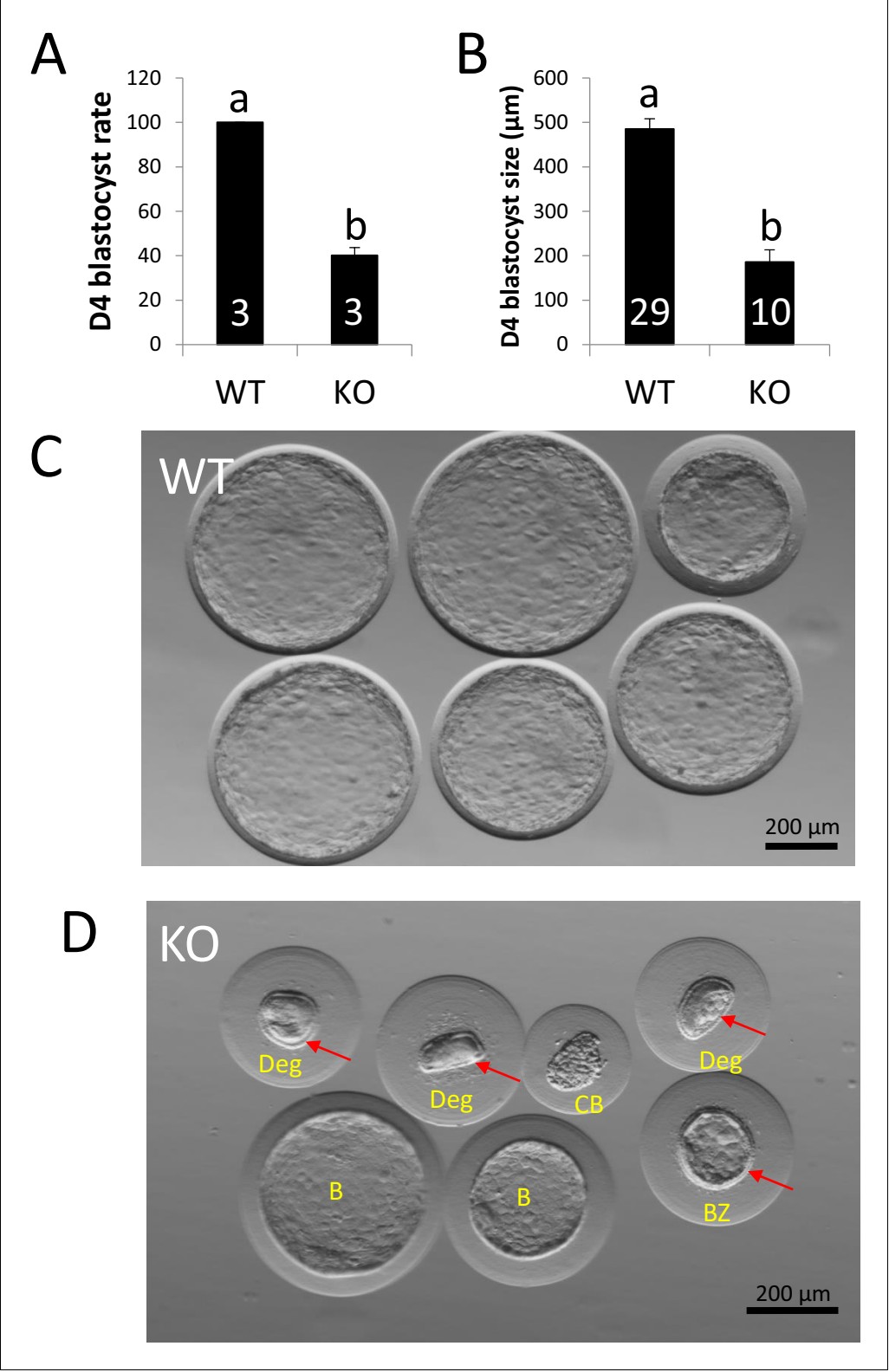

**Figure 3.** ZP4 ablation impairs blastocyst development in vivo. (**A**) Percentage of recovered embryos that had developed to the blastocyst stage by Day 4 post-mating. (**B**) Diameter (µm) of the blastocysts recovered on Day 4. The number of females (**A**) or blastocysts (**B**) analysed per group is indicated inside each column. In both graphs data are represented as mean ± SEM and statistical differences are based on t-test (p<0.05). (**C**) Representative picture of Day four embryos obtained from WT females. (**D**) Representative picture of Day 4 embryos obtained

*Figure 3 continued on next page*

*Figure 3 continued*

from KO females. Red arrows indicate the presence of zona pellucida in some of the structures recovered and yellow letters indicate different types of structures found: Two expanded –yet smaller than those from WT females- blastocysts without zona (B), a collapsed blastocysts (CB) where zona was dissolved displaying a crunched appearance, a smaller expanded blastocyst retaining zona and showing also some degree of mechanical deformation (BZ), and three degenerated embryos (Deg) showing crunched zonae likely due to mechanical damage. Blastomeres (indicating several rounds of cell division before developmental arrest prior to compaction) can be easily spotted on the degenerated embryo on right.

DOI: https://doi.org/10.7554/eLife.48904.007

attachment to the zona matrix to ZP4 in humans (*Chiu et al., 2008*) or to heterocomplexes of ZP4 and ZP3 in porcine and bovine (*Kanai et al., 2007*; *Yurewicz et al., 1998*). In contrast with this notion, the expression of human ZP4 in transgenic mouse ZP was not sufficient to support human sperm binding (*Yauger et al., 2011*). In this article, we have tested ZP4 role by gene ablation in a species where it is naturally present. Although we cannot assure that ZP4 ablation might reduce sperm binding, acrosome reaction or sperm penetration in vitro, our results unequivocally show that ZP4 does not play an essential role during the fertilization process in vivo, but serves a structural and mechanical function which is fundamental to protect the developing embryo prior to implantation. In agreement with our findings, a recent study on ZP characterization hypothesized a structural role of ZP4 on increasing CP thickness in humans (*Nishimura et al., 2019*).

ZP plays an essential protective role during preimplantation embryo development, as despite the exception of a single case report where zona-free embryos were developed in vitro up to blastocyst before transfer (*Shu et al., 2010*), zona denuded embryos have been largely considered unable to complete in vivo development (*Bronson and McLAREN, 1970*; *Modliński, 1970*). This protective function was severely impaired following ZP4 ablation, ultimately causing embryonic death and infertility. ZP4 ablation in rabbits lead to similar structural defects in ZP to those obtained when ZP1 is ablated in mice (*Rankin et al., 1999*): reduction in ZP thickness and increased ZP porosity. However, the effects on fertility of *Zp1* KO were less severe than those caused by ZP4 ablation, as litter size was halved in *Zp1* KO female mice compared to WT mice, whereas ZP4 ablation abolished almost completely embryo development. A plausible explanation for this difference is that mouse embryos may demand a lesser degree of ZP protection during preimplantation development than other domestic mammals, including rabbits. The early blastocyst hatching occurring in mouse (3.5 days compared to more than 6 days in humans and most domestic mammalian species) and a reduced hydrostatic pressure in female reproductive tract motivated by its small body size may be partly responsible for this difference.

Remarkably, the laboratory mouse remain the only mammalian species studied so far where ZP4 is not present in ZP (*Moros-Nicolás et al., 2018*), and mouse ZP is noticeably softer and more elastic compared to other mammalian ZPs containing ZP4 with or without ZP1 (*Yu Sun et al., 2003*). Further evidence for the fundamental structural role of ZP4 comes from transgenic mouse models producing humanized ZP: the expression of human ZP4 in mouse ZP was found to be able to recover the thickness and robustness lost following mouse ZP2 ablation (*Avella et al., 2014*). In order to determine whether ZP4 deficiency could lead to infertility in women, we searched for loss-of-function mutations in the Genome Aggregation Database, which includes data from 141456 individuals (*Lek et al., 2016*). This search revealed 48 polymorphisms in *ZP4* coding region leading to premature stop codons (PTCs), which are the most evident loss-of-function mutations (*Supplementary file 2* and *Figure 6*). The frequency of individuals carrying at least one of these PTC mutations in *ZP4* was 0.3% in the global population analysed, raising to 1% in the African dataset. Hence, in this population, roughly 1 into 40000 women would carry a PTC in both alleles, being thereby potentially infertile due to ZP4 deficiency. Mutations in ZP1-3 genes have been recently associated with women infertility (*Zhou et al., 2019*; *Nishimura et al., 2019*) but the association between these PTC mutations in *ZP4* and woman infertility remains to be explored.

The evolutionary conservation of ZP4 in humans (*Hughes and Barratt, 1999*; *Lefièvre et al., 2004*) and many other mammals (*Goudet et al., 2008*; *Hedrick and Wardrip, 1987*; *Hoodbhoy et al., 2005*; *Izquierdo-Rico et al., 2009*; *Moros-Nicolás et al., 2018*; *Noguchi et al., 1994*; *Stetson et al., 2012*) points towards a conserved role of this protein. In this sense, these

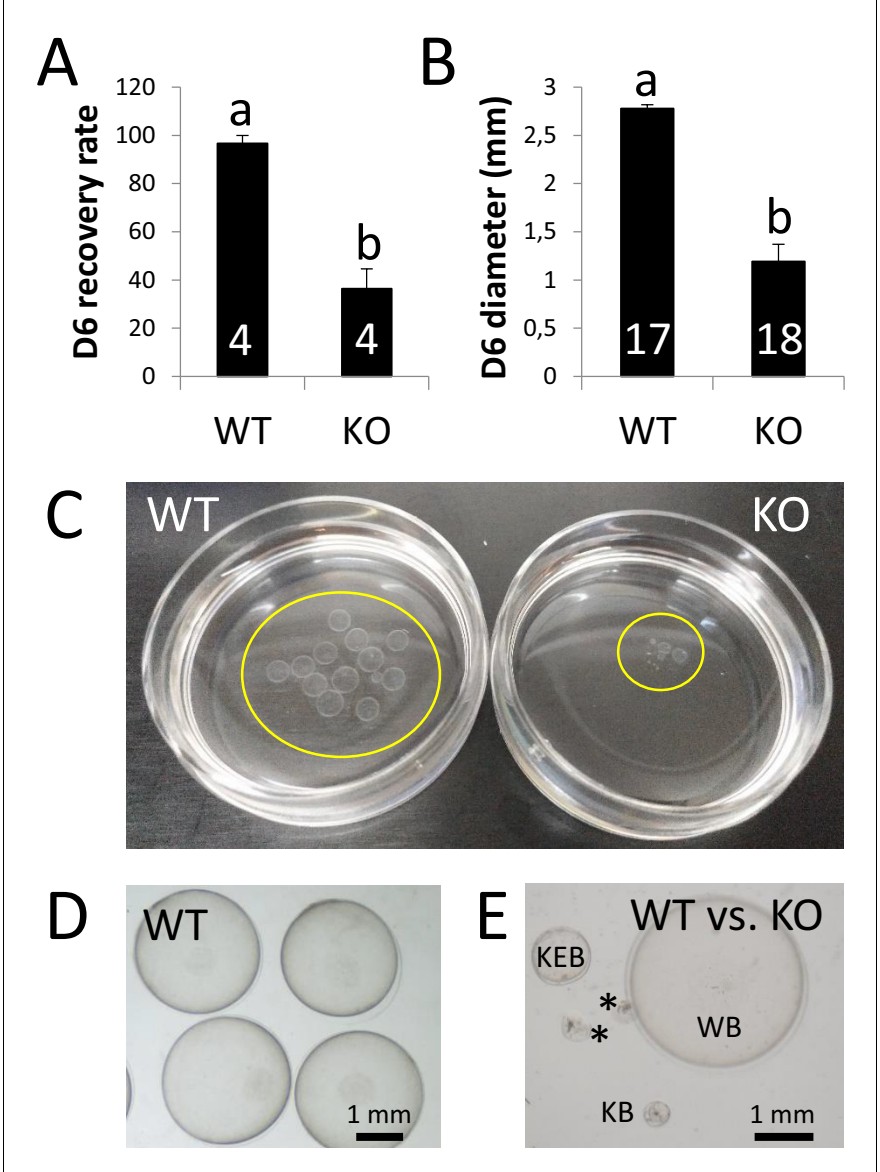

**Figure 4.** ZP4 ablation leads to severely impaired in vivo development to expanded blastocyst. (A) Recovery rate (embryos recovered/corpora lutea) on Day 6 was significantly reduced in KO females compared with WT counterparts. (B) Embryo diameter by Day 6 of in vivo development was significantly inferior in the few embryos recovered from KO females compared to WT counterparts. The number of females (A) or blastocysts (B) analysed per group is indicated inside each column. In both graphs data are represented as mean ± SEM and statistical differences are based on t-test (p<0.05). (C) Representative picture of Day 6 embryos recovered from a WT (left) or KO (right) female. Embryos were placed on 35 mm dishes. Yellow circles mark the location of the embryos inside the dishes. (D–E) Representative pictures of Day 6 embryos recovered from WT (D) or WT and KO (E) females. (E) 'WB' is a Day 6 blastocyst from the WT group, whereas the other structures were recovered from a KO female: KEB is an expanded blastocyst, KB a blastocyst showing a lesser degree of expansion and asterisks mark two degenerated embryos.

DOI: https://doi.org/10.7554/eLife.48904.008

results highlight *ZP4* mutations as a possible cause for female infertility in both humans and livestock or wild species, and pave the way for the development of contraceptive methods based on ZP4 disruption.

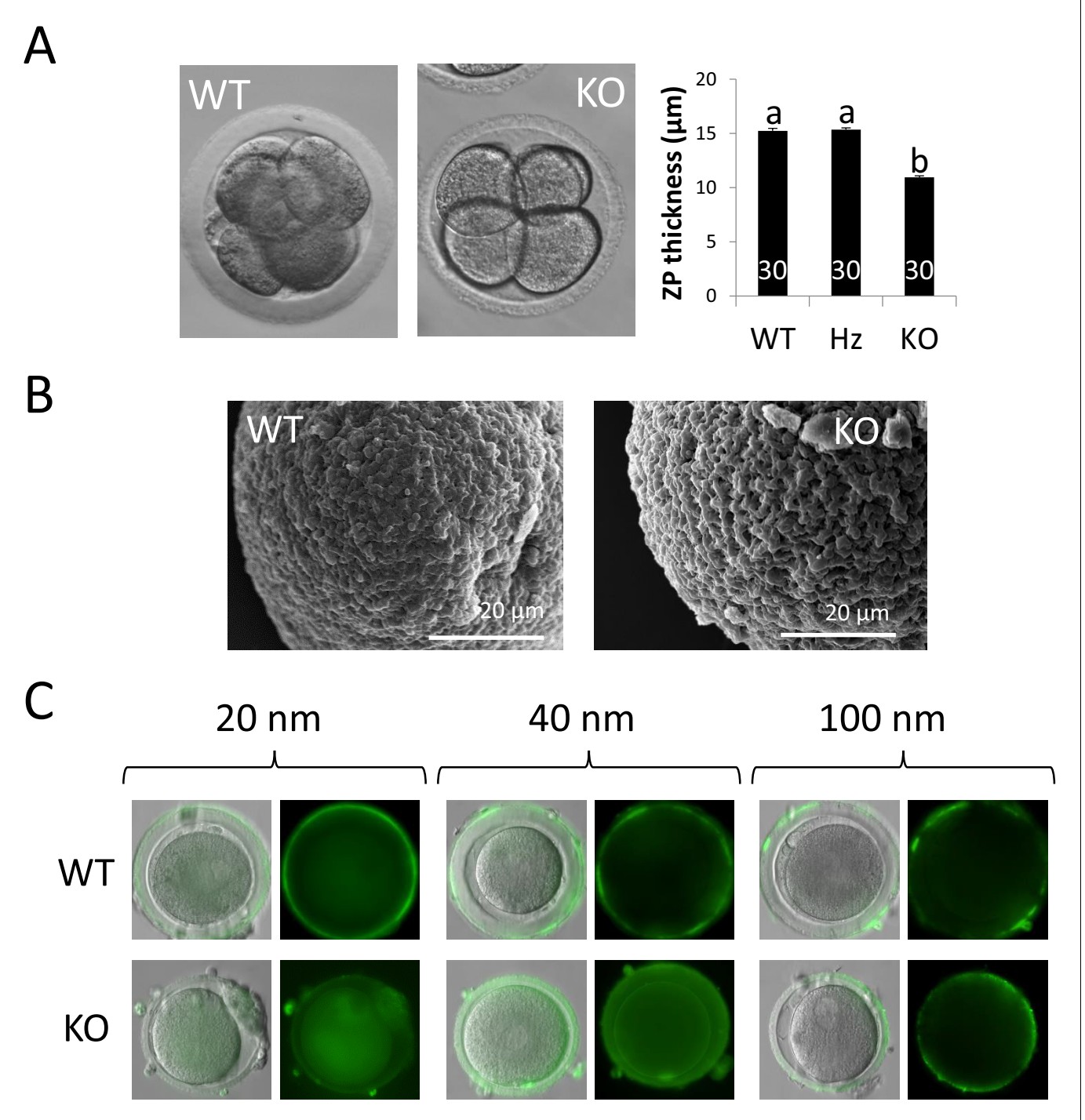

**Figure 5.** ZP4-devoided zona show structural abnormalities. (**A**) Representative pictures of cleaved embryos enclosed in ZP with (left) or without (right) ZP4 protein. ZP lacking ZP4 was significantly thinner than WT ZP. Hz females produced ZP overtly undistinguishable from WT counterparts. Data are represented as mean ± SEM and statistical differences are based on ANOVA (p<0.05). (**B**) Surface Electron Microscopy (SEM) pictures of ZP with (left) or without (right) ZP4 protein. In the absence of ZP4, ZP texture looks rougher and more fenestrated. (**C**) Fluorescent nanospheres permeability test in zygotes surrounded by ZP with (upper raw) or without (lower raw) ZP4 protein. WT and KO ZP blocked 100 nm particles, as fluorescence was restricted to the outer surface of ZP in both groups (right images). In contrast, all 9 ZP4-devoided ZP analyzed were permeable to 20–40 nm nanospheres, as green fluorescence inside ZP was evident (left and central images). No green fluorescence can be observed inside WT ZP, indicating that 20–40 nm nanospheres were efficiently blocked by a ZP containing ZP4. Statistical differences based on Chi-square test (p<0.05).

DOI: https://doi.org/10.7554/eLife.48904.009

# Materials and methods

## Key resources table

| Reagent type (species) or resource | Designation | Source or reference | Identifiers | Additional information |
|---|---|---|---|---|
| Strain, strain background (*Oryctolagus cuniculus*) | *ZP4* KO | This article | | Generated by CRISPR (M and M), available upon request. |
| Antibody | Goat polyclonal anti-human ZP4 | Santa Cruz | #sc-49586 RRID:AB2304934 | (1:2000) |
| Antibody | ImmPRESS Horse anti-goat IgG HRP | Vector Labs | #MP-7405 RRID:AB2336526 | (1X, ready-to-use reagent) |
| Recombinant DNA reagent | Plasmid pMJ920 | Addgene | #42234 | *Jinek et al., 2013* |
| R ecombinant DNA reagent | Plasmid px330 | Addgene | #42230 | *Yang et al., 2014* |
| Sequenced-based reagent | Primers | This article | | *Table 1* (ordered from Sigma) |

## Animal models

All animal protocols were approved by INIA Animal Welfare Committee and Madrid Region authorities (authorization PROEX040/17). Rabbits (*Oryctolagus cuniculus*, farm commercial hybrids of New Zealand White) were housed individually maintaining visual and olfactory contact with others to allow social interactions. Temperature was controlled to 20–24°C and light cycle was 14:10.

The generation of Type I genetically modified *O. cuniculus* was approved by Spanish National Biosafety Agency (authorization A/ES/17/03). *ZP4* KO were generated by CRISPR/Cas system. A single guide RNA (sgRNA) was designed at the beginning of the coding region of rabbit *ZP4* gene using MIT CRISPR design tool (*Yang et al., 2014*), which also provided the five most probable off-targets. sgRNA and capped polyadenylated Cas9 mRNA were synthesized in vitro as described previously (*Bermejo-Álvarez et al., 2015*). Briefly, capped polyadenylated Cas9 mRNA was produced by in vitro transcription (mMESSAGE mMACHINE T7 ULTRA kit, Life Technologies) from the plasmid pMJ920 (Addgene 42234) linearized with BstBI. sgRNA was produced by in vitro transcription (MEGAshortscript T7 kit, Life Technologies) from a PCR-amplified template using px330 vector (Addgene 42230) as previously described (*Yang et al., 2014*). Cas9 mRNA and sgRNA were co-injected at 100 ng/µl and 25 ng/µl, respectively, into rabbit zygotes collected by oviductal flushing 14 hr after natural mating. Microinjected embryos were transferred one day after microinjection to the oviduct of a pseudopregnant recipient stimulated by a 0.02 mg gonadorelin IM injection (Inducel, Laboratorios Ovejero) on the previous day. Resulting pups were genotyped by clonal sequencing as previously described (*Bermejo-Álvarez et al., 2015*). Genotyping was performed on both ear biopsies and blood samples in F0 generation. Following DNA purification, the genomic sequence surrounding CRISPR target site was amplified by PCR and the purified PCR product was cloned into pMD20 vector (Takara) and transformed into competent cells. 15 clones were sequenced to detect the unknown alleles (indels) generated by NHEJ-repair of

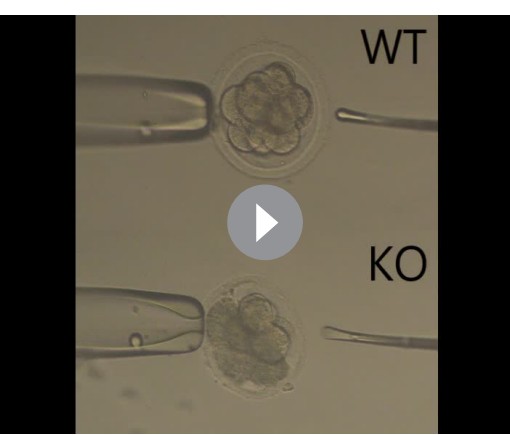

**Video 1.** WT and KO ZP exhibit different behavior when mechanical pressure is exerted by a blunt micromanipulation pipette. WT ZP (upper embryo) handles easily the pressure quickly recovering its spherical shape. In contrast, ZP4-devoided ZP (lower embryo) is easily deformable, providing a poor protection to the embryo.
DOI: https://doi.org/10.7554/eLife.48904.010

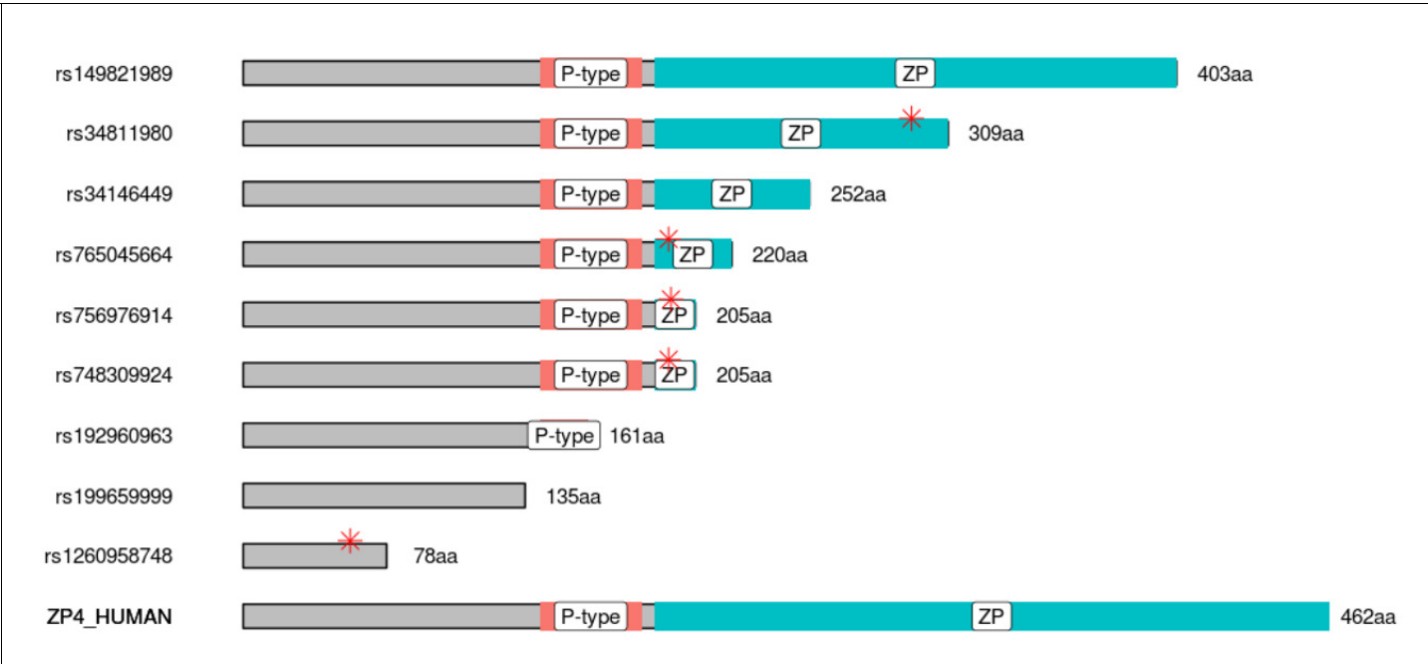

**Figure 6.** Polymorphism leading to truncated human ZP4 protein can be found at gnomAD database. Graphical representation of the human truncated proteins generated by the nine most frequent premature stop codon (PTCs) polymorphisms found in gnomAD database. Orange bars depicts P-type (trefoil domain), blue bars correspond to ZP domain, and red asterisks mark the position of the first amino acid affected by frame shift mutation (i.e., the end of the WT protein sequence).

DOI: https://doi.org/10.7554/eLife.48904.011

CRISPR-induced DSBs on each individual. In subsequent generations, Sanger PCR sequencing was performed, as all parental alleles were already known, allowing its identification in a mixed sequencing reaction. Details of primers used for sgRNA synthesis, genotyping and off-target analysis are provided in *Table 1*.

**Table 1.** Details of primers used for sgRNA cloning into px330 vector (*sgZP4*, target sequence in bold), PCR template generation for sgRNA synthesis (*T7sgZP4*), and genotyping of target (*GenoZP4*) and off-target sequences (*OffZP4 1–5*).

| Primer | Primer sequence (5´–3´) | Fragment size (bp) | GenBank accession no. |
|---|---|---|---|
| *sgZP4* | CACCG**GTAGCACCATGTGGCTGTTA** AAACTAACAGCCACATGGTGCTACC | - | NC_013684.1 |
| *T7sgZP4* | GATCACTAATACGACTCACTATAGGG-TGCTGTCAGGGAGCTCAGTTTTAGAG-CTAGAAAT | - | - |
| *GenoZP4* | TGTTGCCCAACTTACCCCAG TCCCACCTGCATGCAAGAAA | 318 | NC_013684.1 |
| *OffZP4 1* | CAGCTGCAGTGGAAGAATGC ACTGTAACGCAGCTTTGGGA | 498 | NW_003159362.1 |
| *OffZP4 2* | AACAGACACCCAGAAGCACC ACTGCCTTTTCCCACAAGCA | 234 | NC_013699.1 |
| *OffZP4 3* | TCTCATCAGAGCTCCGTCCA CTGGGAAACCACTCTCCGTC | 395 | NW_003159579.1 |
| *OffZP4 4* | ACAACTGTTTCAGCTAACCTCT GTTGAGTGGTCACCAAGCCT | 313 | NC_013678.1 |
| *OffZP4 5* | GCCCAGGGGTTAACACACAGT TCCTCTTTGACCCCTCTCCC | 264 | NC_013686.1 |

DOI: https://doi.org/10.7554/eLife.48904.012

## Immunohistochemistry

Immunohistochemistry was performed on ovaries (3/group) collected 14 hr after mating, fixed in 10% formaldehyde for 24 hr, processed and paraffin embedded. Rehydrated sections of 5 µm were incubated in 0.3% hydrogen peroxide in TBS to block endogenous peroxidase activity and then in primary antibody solution (goat polyclonal anti-human ZP4 (Santa-Cruz sc-49586) dilution 1:2000 in TBS-1% BSA solution) for 1 hr at room temperature in a moist chamber. After a brief wash in TBS, sections were incubated in secondary antibody solution (ImmPRESS horse anti-goat IgG HRP, Vector Labs MP-7405) for 30 min at 37°C and immunoreaction was revealed with 3–3′diaminobencidine. Finally, sections were counterstained with Harry´s hematoxylin (Thermo Scientific), dehydrated, cleared and mounted. Positive immunoreaction was identified as a dark-brown precipitated. The images were collected with a Leica DM 6000 microscope and Software Leica Application Suite.

## Proteomics analyses

Proteomics was performed on 50 grown oocytes/group recovered by slicing ovaries in PBS. ZP was solubilized by incubation at 75°C for 45 min (*Izquierdo-Rico et al., 2009*) and analyses were carried out on one technical replicate with a HPLC/MS system consisting of an Agilent 1290 Infinity II Series HPLC (Agilent Technologies, Santa Clara, CA, USA) equipped with an Automated Multisampler module and a High Speed Binary Pump, and connected to an Agilent 6550 Q-TOF Mass Spectrometer (Agilent Technologies, Santa Clara, CA, USA) using an Agilent Jet Stream Dual electrospray (AJS-Dual ESI) interface. Experimental parameters for HPLC and Q-TOF were set in MassHunter Workstation Data Acquisition software (Agilent Technologies, Rev. B.08.00). Data processing and analysis was performed on Spectrum Mill MS Proteomics Workbench (Rev B.06.00.201, Agilent Technologies, Santa Clara, CA, USA).

## Fertility analyses

All experiments were performed following natural mating with WT males of proven fertility. Ovulation in rabbits is induced by mating, allowing precise determination of embryo developmental timing. Zygotes were collected post-mortem 14 hr after mating by oviductal flushing with Dulbecco´s PBS media supplemented with 1% BSA (DPBS) in five independent replicates. Ovulation rates were assessed by counting corpora lutea. Embryo culture took place on 25 µl drops of TCM-199 media (Sigma) supplemented with 5% FCS at 38.5°C in a 5% $CO_2$ and 5% $O_2$ water saturated atmosphere. Embryonic cleavage and development to blastocysts were assessed on Days 1 and 5 of culture, respectively.

Follicular counts were performed in histological sections of ovaries collected 14 hr after mating, fixed and stained with Haematoxylin-Eosin as detailed above (four sections per ovary, four animals per group). Pronuclear formation rates were analysed in the zygotes obtained from three females/group fixed 20 hr following mating as described previously (*Bermejo-Alvarez et al., 2012*). Pronuclei were stained by incubation in 1 µM SYTOX Green (Thermofisher) for 30 min and observed under fluorescent inverted microscope (Nikon Eclipse).

To analyse in vivo development to Day 4 blastocyst, three females/group were sacrificed 4 days after mating and oviducts (where no structures were recovered) and uteri were independently flushed with DPBS. The number of structures recovered were counted and blastocysts were observed and measured under light estereomicroscopy (Zeiss V20 coupled with Hammamatsu Orca Flash 4.0 camera). A similar approach was followed to analyse in vivo development to Day 6 blastocysts, but in this case four females/group were used and only uteri were flushed. Corpora lutea and the number of structures were counted to determine recovery rates. The structures recovered were observed and measured under light estereomicroscopy (Zeiss Stemi 305 coupled with Izasa CMOS 1080P camera).

## Structural analyses

Zona pellucida thickness was analysed in 30 fresh embryos/group 40–43 hr post-mating by inverted light microscopy (Nikon Eclipse) using the measurement tool of the software NIS (Nikon).

Scanning Electron Microscopy (SEM) was performed on five zygotes/group fixed in 2% glutaraldehyde for 2 hr at 4°C, washed in PBS and postfixed in 1% osmium tetroxide for 1 hr. After washing,

the zygotes were dehydrated in increasing concentration of acetone and air dried. Finally, samples were sputter coated with gold and studied in Jeol-6100 scanning electron microscope.

ZP permeability test was performed by incubating fresh zygotes (3 replicates of 3 embryos/group) in 0.005% (1:1000 dilution) solutions of carboxylate-modified 505/515 Fluospheres (Invitrogen) of different sizes (20, 40 and 100 nm) for 30 min, using the culture media and atmosphere conditions detailed above. Immediately following incubation and washing, zygotes were observed under an inverted fluorescent microscope to assess the penetration of fluorescent particles. This test yields a binary result: green fluorescence can be detected inside ZP (ZP is permeable to that size of particle) or fluorescence cannot be detected inside ZP (particles do pass through ZP).

### Genomic analysis

Loss of function ZP4 polymorphisms passing QC filters were queried online at (http://gnomad. broadinstitute.org/gene/ENSG00000116996), and a cvs file was downloaded and manipulated with R v3.5.1. *Figure 6* was plotted taking as reference the protein sequence UniProt Q12836 and using the packages ggplot2 and drawproteins (*Brennan, 2018*), which was customized to depict frameshift mutations. Biosequence analysis using profile hidden Markov models was performed online with HMMER v1.32 (*Potter et al., 2018*).

### Statistical analyses

The differences between groups in litter size, ovulation and developmental rates, follicle number, in vivo embryo survival and ZP and embryo size were analysed by One-way ANOVA following Tukey´s post-hoc test or by t-test to compare 3 or two groups, respectively, except for permeability test (binary variable), where Chi-square test were used. All statistical tests were performed using the software package SigmaStat. Differences were considered statistically significant at $p < 0.05$.

## Acknowledgements

This work was supported by Grants AGL2014-58739-R, RYC-2012-10193, AGL2017-84908R, AGL2016-71890-REDT, PGC2018-094781-B-100, AGL2015-70159-P and AGL2015-65572-C2-1-R from the Spanish Ministries of Economy and Competitiveness and Science, Innovation and University, and 757886-ELONGAN from the European Research Council. ILT and NFB are supported by FPI fellowships from the Spanish Ministry of Economy and Competitiveness. AQF by Marie Curie Action FP7/2007-2013 grant 600391 from EU, and LGB by a FPU fellowship from the Spanish Ministry of Education, Culture and Sports.

## Additional information

### Funding

| Funder | Grant reference number | Author |
| --- | --- | --- |
| Ministerio de Economía y Competitividad | AGL2014-58739-R | Pablo Bermejo-Álvarez |
| Ministerio de Economía y Competitividad | RYC-2012-10193 | Pablo Bermejo-Álvarez |
| Ministerio de Economía y Competitividad | AGL2017-84908R | Pablo Bermejo-Álvarez |
| Ministerio de Economía y Competitividad | AGL2016-71890-REDT | Pedro L Lorenzo Manuel Avilés Pablo Bermejo-Álvarez |
| Ministerio de Economía y Competitividad | AGL2015-70159-P | Manuel Avilés |
| Ministerio de Economía y Competitividad | AGL2015-65572-C2-1-R | Pedro L Lorenzo Pilar García-Rebollar |
| Ministerio de Economía y Competitividad | FPI fellowships | Ismael Lamas-Toranzo Noelia Fonseca Balvís |

| H2020 European Research Council | 757886-ELONGAN | Pablo Bermejo-Álvarez |
| European Union Seventh Framework Programme | FP7/2007-2013 600391 | Ana Querejeta-Fernández |
| Ministerio de Educación, Cultura y Deporte | FPU fellowship | Leopoldo González-Brusi |
| Ministerio de Ciencia, Innovación y Universidades | PGC2018-094781-B-100 | Manuel Avilés |

The funders had no role in study design, data collection and interpretation, or the decision to submit the work for publication.

## Author contributions

Ismael Lamas-Toranzo, Formal analysis, Investigation, Methodology, Writing—review and editing, Participated in KO generation, performed all reproductive and developmental analyses and genotyping and was involved in permeability test; Noelia Fonseca Balvís, Investigation, Methodology, Participated in KO generation; Ana Querejeta-Fernández, Resources, Supervision, Investigation, Methodology, Writing—review and editing, Involved in permeability test; María José Izquierdo-Rico, Investigation, Methodology, Writing—review and editing, Responsible for SEM and proteomics analyses; Leopoldo González-Brusi, Formal analysis, Investigation, Methodology, Writing—review and editing, Responsible for SEM and proteomics analyses and performed human genomic analysis; Pedro L Lorenzo, Supervision, Funding acquisition; Pilar García-Rebollar, Resources, Supervision, Funding acquisition, Project administration; Manuel Avilés, Conceptualization, Resources, Formal analysis, Supervision, Funding acquisition, Investigation, Methodology, Writing—original draft, Project administration, Writing—review and editing, Responsible for SEM and proteomics analyses; Pablo Bermejo-Álvarez, Conceptualization, Resources, Formal analysis, Supervision, Funding acquisition, Validation, Investigation, Methodology, Writing—original draft, Project administration, Writing—review and editing, Participated in KO generation and performed all reproductive and developmental analyses and genotyping

## Author ORCIDs

Ismael Lamas-Toranzo (ID) https://orcid.org/0000-0002-7790-2649
Pablo Bermejo-Álvarez (ID) https://orcid.org/0000-0001-9907-2626

## Ethics

Animal experimentation: All animal protocols were approved by INIA Animal Welfare Committee and Madrid Región authorities (authorization PROEX040/17).

## Decision letter and Author response

Decision letter https://doi.org/10.7554/eLife.48904.017
Author response https://doi.org/10.7554/eLife.48904.018

# Additional files

## Supplementary files

• Supplementary file 1. Tandem mass spectrometry proteomics analysis of WT or *ZP4* KO solubilized ZPs showing the presence of all ZP proteins but ZP4 in *ZP4* KO ZPs.
DOI: https://doi.org/10.7554/eLife.48904.013

• Supplementary file 2. Polymorphism leading to premature stop codons (PTCs) found in gnomAD database, organized in order of decreasing allele count.
DOI: https://doi.org/10.7554/eLife.48904.014

• Transparent reporting form
DOI: https://doi.org/10.7554/eLife.48904.015

## Data availability

All data generated or analyzed during this study are included in the manuscript and supporting files.

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
