## [Decision Letter]

Thank you for submitting your article "ZP4 confers structural properties to the zona pellucida essential for embryo development" for consideration by *eLife*. Your article has been reviewed by three peer reviewers, and the evaluation has been overseen by a Reviewing Editor and Michael Eisen as the Senior Editor. The following individual involved in review of your submission has agreed to reveal their identity: Véronique Duranthon (Reviewer #2).

The reviewers have discussed the reviews with one another and the Reviewing Editor has drafted this decision to help you prepare a revised submission.

The zona pellucida is an essential oocyte structure for fertilization and subsequent embryonic development. Yet, the composition of this structure is species-specific, hence the difficulty to make assumptions from one animal model to the other, and to humans ultimately. ZP4 is a glycoproteic component of the zona pellucida that exists in most eutherian mammals, except the mouse. The authors report here the first ablation (CRISPR-Cas9-driven) of the Zp4 gene in rabbits and its effect on development. The zona pellucida structure appears affected in absence of ZP4, presenting as thinner and more permeable compared to wildtype. Interestingly, compared to other Zp genes, ZP4 is not absolutely required for oocyte fertilization, but has post-fertilization effects in vivo and is essential for female fertility, while being dispensable for in vitro preimplantation development.

The reviewers agreed on the fact that this is an important work, with novel conclusions and interesting implication for humans. However, there was also consensus towards the conclusion that more work is needed to properly delineate the role of ZP4 for the embryo and that some of the conclusions are not fully supported by the data..

The reviewers are well aware that obtaining edited rabbits is challenging, so between essential points 1 and 2, a choice may have to be made, regarding the limited period of 2 months that *eLife* allows for revision. Personally, I think point 2 may be the more interesting – if equally feasible – to experimentally address, as it may point to the reason why embryonic development fails in absence of ZP4. But wherever experiments could not be carried out, conclusions should be toned down, according to the recommendations below.

Essential revisions:

1) The lack of effect of the ZP4 mutation on the process of fertilization is not directly proven, only implied. The authors should be more cautious in their conclusion. Additional experimental work would be needed to exclude any effect on fertilization-related events per se, like sperm binding with the egg, acrosome reaction, sperm penetration etc. There are precedents in the literature of fertilization-related proteins whose role can be masked in vivo due to compensatory mechanisms. If more edited females can be obtained, in vitro fertilization needs to be performed and sperm binding, penetration, fertilization, and polyspermy quantified under clearly-defined conditions. If it is not feasible, then the conclusion that the fertilization step is not affected should be attenuated, by limiting the conclusion to a non-oriented statement of the results = cleavage is not affected and therefore ZP4 is not absolutely required for fertilization in vivo.

2) The discrepancy between the in vivo and in vitro developmental outcome should be better addressed. Because blastocysts were lost at Day 6 of in vivo development, the authors concluded that the main function of ZP4 is to protect the preimplantation embryo. However, this is clearly not proven. If the authors want to prove that the phenotype at Day 6 is due to preimplantation loss, further investigations are required to show when the loss occurs and establish the phenotype of that loss. Moreover, the authors fail to mention the dynamics of the rabbit zona pellucida, which disappears in vivo between Day 3 and Day 4 to be replaced by a neozona (Denker et al., 1979). Such a modification does not occur in the in vitro culture conditions used in the paper (Fisher et al., 1991). Additionally, during embryo transit in the oviduct, a mucin coat covers the developing embryo and is replaced by a gloiolemma at Day 5-6. The lack of development observed in vivo could equally result from effects of ZP4 on the formation of the mucinous coat, or, on the replacement of the zona by the neozona, none of these changes of the extracellular matrix being recapitulated in vitro. Whether ZP4 mutation may have rabbit specific effects due to these features should be discussed.

3) About the part on human genetics of the ZP4 gene: the authors found 48 polymorphisms in ZP4 coding region leading to premature stop codons in human. They claim that ZP4 deficiencies could thus lead to infertility in human. As such, this conclusion seems overstated: only an enrichment for such mutations in infertile patients may have led to such a conclusion. This paragraph (subsection “ZP4-devoided zona show structural abnormalities”, last paragraph) should be moved in the Discussion section.

4) Statistical and quantitative analyses should be strengthened, information is often missing or incomplete.

- mention on each graph the numbers of embryos observed in each category and not only the percentages obtained (Figure 2C, 2D, 2G, 3A, 3B, 4A).

- Quantitative assessments on ovaries are needed to conclude that there was no effect on the development of oocytes.

- Same for the analysis of embryonic development. For example, was there any effect on inner cell mass development?

- Figure 2: Please mention in the legend the number of females involved. This is only mentioned in the Materials and methods section, which makes reading uneasy.

- Only a single example of the nanosphere experiment is shown, and no quantification is reported. A quantitative, rigorous analysis is needed, with statistical tests.

- The statistical analysis section of Materials and methods says analyses were done by ANOVA. However, ANOVA is not appropriate for only two groups (Figure 3), where a t-test should be used to compare means. ANOVA alone also cannot show comparisons among the treatment groups. A post-hoc test (e.g., Tukey's or Dunnet's) is needed. This should be specified. Also, was it confirmed that the data were normally distributed and ANOVA was appropriate? This seems unlikely for proportion data like in Figure 2.

---

## [Author Response]

Essential revisions:1) The lack of effect of the ZP4 mutation on the process of fertilization is not directly proven, only implied. The authors should be more cautious in their conclusion. Additional experimental work would be needed to exclude any effect on fertilization-related events per se, like sperm binding with the egg, acrosome reaction, sperm penetration etc. There are precedents in the literature of fertilization-related proteins whose role can be masked in vivo due to compensatory mechanisms. If more edited females can be obtained, in vitro fertilization needs to be performed and sperm binding, penetration, fertilization, and polyspermy quantified under clearly-defined conditions. If it is not feasible, then the conclusion that the fertilization step is not affected should be attenuated, by limiting the conclusion to a non-oriented statement of the results = cleavage is not affected and therefore ZP4 is not absolutely required for fertilization in vivo.

We apologize for not being able to provide IVF data. Rabbit sperm capacitation is difficult to achieve in vitro and, hence, rabbit IVF is particularly challenging. We routinely perform IVF in different mammalian species (bovine, mouse, sheep and pig), but to perform rabbit IVF we would need to develop a specific protocol which would not be ready to use in a reasonable time. As a result, we have toned down our conclusions by stating that these results prove that ZP4 is dispensable for fertilization in vivo (subsection “ZP4 ablation leads to severely impaired fertility” and Discussion), as similar pronuclear formation (i.e., similar monospermic sperm penetration) and embryonic cleavage (oocyte activation following sperm penetration) rates were obtained following in vivofertilization of WT and KO females (Figure 2 C, G and H).

2) The discrepancy between the in vivo and in vitro developmental outcome should be better addressed. Because blastocysts were lost at Day 6 of in vivo development, the authors concluded that the main function of ZP4 is to protect the preimplantation embryo. However, this is clearly not proven. If the authors want to prove that the phenotype at Day 6 is due to preimplantation loss, further investigations are required to show when the loss occurs and establish the phenotype of that loss. Moreoever, the authors fail to mention the dynamics of the rabbit zona pellucida, which disappears in vivo between Day 3 and Day 4 to be replaced by a neozona (Denker et al., 1979). Such a modification does not occur in the in vitro culture conditions used in the paper (Fisher et al., 1991). Additionally, during embryo transit in the oviduct, a mucin coat covers the developing embryo and is replaced by a gloiolemma at Day 5-6. The lack of development observed in vivo could equally result from effects of ZP4 on the formation of the mucinous coat, or, on the replacement of the zona by the neozona, none of these changes of the extracellular matrix being recapitulated in vitro. Whether ZP4 mutation may have rabbit specific effects due to these features should be discussed.

We agree that the failure on in vivodevelopment required further characterization. To exclude any possible effect attributable to the formation of neozona or to mucin coat substitution, we have recovered embryos fertilized and developed in vivoon Day 4. At that stage the original zona pellucida has disappeared and the embryo is only surrounded by mucin coat, as neozona will be formed later (Day 5). As it can be seen in the new Figure 3, embryos derived from WT or KO females were covered by mucin and ZP was dissolved in the developing embryos surrounded by KO ZP. ZP was present in degenerated embryos as ZP dissolution requires from embryo development (Fisher et al., 1991). However despite mucin coating was normal and ZP without ZP4 could be dissolved, a clear reduction in development to blastocyst and in blastocysts size can be already noticed by that day (i.e. before mucin-gloiolemma exchange and neozona formation). Degenerated embryos and even some blastocyst trying to expand show a crunched appearance compatible with a mechanical damage occurring in the absence of ZP4. We have included a new figure with these results (Figure 3) and have modified the Results section (subsection “ZP4 ablation leads to severely impaired fertility”).

3) About the part on human genetics of the ZP4 gene: the authors found 48 polymorphisms in ZP4 coding region leading to premature stop codons in human. They claim that ZP4 deficiencies could thus lead to infertility in human. As such, this conclusion seems overstated: only an enrichment for such mutations in infertile patients may have led to such a conclusion. This paragraph (subsection “ZP4-devoided zona show structural abnormalities”, last paragraph) should be moved in the Discussion section.

We have moved the paragraph to the Discussion section (third paragraph) and stated more clearly that we have not proved the association between the PTC mutations identified by our analysis in *ZP4* with women infertility.

4) Statistical and quantitative analyses should be strengthened, information is often missing or incomplete.- Mention on each graph the numbers of embryos observed in each category and not only the percentages obtained (Figure 2C, 2D, 2G, 3A, 3B, 4A).

We apologize for not providing earlier that information in the figures, it was included in the *eLife*´s transparent reporting form, but it will be definitely more accessible to the readers in the columns. We have now included n in all columns of the figures and detailed its meaning on figure legends.

- Quantitative assessments on ovaries are needed to conclude that there was no effect on the development of oocytes.

We have performed follicular counts on 4 females per group analysing 4 sections per ovary (16 sections/group). Similar numbers of antral and preantral follicles were obtained (Figure 2E-F).

- Same for the analysis of embryonic development. For example, was there any effect on inner cell mass development?

in vitro produced blastocysts were morphologically similar in both groups, and all blastocysts develop ICM. However, in vivodevelopment was impaired: more than half of the embryos were not able to develop to blastocyst on Day 4. Those reaching the blastocyst stage in the KO group do develop ICM, but those ICM were smaller, as the rest of the embryo. Later on development (Day 6) the developmental impairment of the KO group was more evident, but again, those embryos developing to slightly expanded blastocyst do show embryonic disc, although it was less developed than that found in the WT group (as the rest of the embryo). In other words, it seems that the damage did not affect preferentially the ICM: both TE and ICM development were reduced when ZP4 was absent from ZP.

- Figure 2: Please mention in the legend the number of females involved. This is only mentioned in the Materials and methods section, which makes reading uneasy.

We have added this information to the figure.

- Only a single example of the nanosphere experiment is shown, and no quantification is reported. A quantitative, rigorous analysis is needed, with statistical tests.

We should have provided more details for this test. This assay yields a binary result: green fluorescence detected inside ZP (permeable) or no fluorescence detected inside ZP (not permeable). All WT embryos (9) analysed following incubation with 40 or 20 nm nanospheres lacked fluorescence inside the ZP and all KO embryos (9) showed fluorescence. We have modified the subsections “ZP4-devoided zona show structural abnormalities” and “Structural analyses” to explain better this test.

- The statistical analysis section of Materials and methods says analyses were done by ANOVA. However, ANOVA is not appropriate for only two groups (Figure 3), where a t-test should be used to compare means. ANOVA alone also cannot show comparisons among the treatment groups. A post-hoc test (e.g., Tukey's or Dunnet's) is needed. This should be specified. Also, was it confirmed that the data were normally distributed and ANOVA was appropriate? This seems unlikely for proportion data like in Figure 2.

We have now performed t-test when 2 groups are compared. Regarding to ANOVA, the statistical software we use automatically perform ANOVA on ranks followed by Tukey´s when normalcy test fail, as it was the case for pregnancy data (Figure 2A). We have modified figure legend and Materials and methods accordingly.